# Variational Mixture-of-Experts Autoencoders for Multi-Modal Deep Generative Models

**Yuge Shi**[*]      **N. Siddharth**[*]
Department of Engineering Science
University of Oxford
`{yshi, nsid}@robots.ox.ac.uk`

**Brooks Paige**
Alan Turing Institute &
University of Cambridge
`bpaige@turing.ac.uk`

**Philip H.S. Torr**
Department of Engineering Science
University of Oxford
`philip.torr@eng.ox.ac.uk`

## Abstract

Learning generative models that span multiple data modalities, such as vision and language, is often motivated by the desire to learn more useful, generalisable representations that faithfully capture common underlying factors between the modalities. In this work, we characterise successful learning of such models as the fulfilment of four criteria: i) implicit latent decomposition into shared and private subspaces, ii) coherent joint generation over all modalities, iii) coherent cross-generation across individual modalities, and iv) improved model learning for individual modalities through multi-modal integration. Here, we propose a mixture-of-experts multimodal variational autoencoder (MMVAE) to learn generative models on different sets of modalities, including a challenging image ↔ language dataset, and demonstrate its ability to satisfy all four criteria, both qualitatively and quantitatively. Code, data, and models are provided at this url.

## 1 Introduction

Human learning in the real world involves a multitude of perspectives of the same underlying phenomena, such as perception of the same environment through visual observation, linguistic description, or physical interaction. Given the lack explicit labels available for observations in the real world, observing *across modalities* can provided important information in the form of correlations between the observations. Studies have provided evidence that the brain jointly embeds information across different modalities (Quiroga et al., 2009; Stein et al., 2009), and that such integration benefits reasoning and understanding through expression along these modalities (Bauer and Johnson-Laird, 1993; Fan et al., 2018), further facilitating information transfer between (Yildirim, 2014) them. We take inspiration from this to design algorithms that handle such multi-modal observations, while being capable of a similar breadth

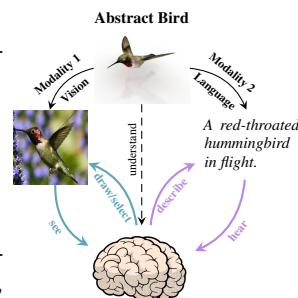

Figure 1: A schematic for multi-modal perception.

of behaviour. Figure 1 shows an example of such a situation, where an abstract notion of a bird is perceived through both visual observation as well as linguistic description. A thorough understanding of what a bird is involves understanding not just the characteristics of its visual and linguistic features individually, but also how they relate to each other (Barsalou, 2008; Siskind, 1994). Moreover, demonstrating such understanding involves being able to visualise, or discriminate birds against other things, or describe birds' attributes. Crucially, this process involves flow of information in *both* ways—from observations to representations and vice versa.

---

[*]Equal contribution

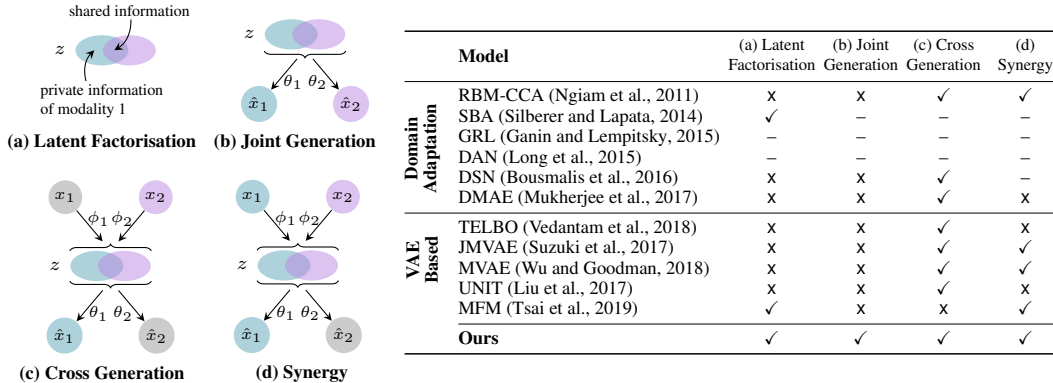

**(a) Latent Factorisation**  **(b) Joint Generation**

**(c) Cross Generation**  **(d) Synergy**

| | Model | (a) Latent Factorisation | (b) Joint Generation | (c) Cross Generation | (d) Synergy |
|---|---|---|---|---|---|
| **Domain Adaptation** | RBM-CCA (Ngiam et al., 2011) | x | x | ✓ | ✓ |
| | SBA (Silberer and Lapata, 2014) | ✓ | – | – | – |
| | GRL (Ganin and Lempitsky, 2015) | – | – | – | – |
| | DAN (Long et al., 2015) | – | – | – | – |
| | DSN (Bousmalis et al., 2016) | x | x | ✓ | – |
| | DMAE (Mukherjee et al., 2017) | x | x | ✓ | x |
| **VAE Based** | TELBO (Vedantam et al., 2018) | x | x | ✓ | x |
| | JMVAE (Suzuki et al., 2017) | x | x | ✓ | ✓ |
| | MVAE (Wu and Goodman, 2018) | x | x | ✓ | ✓ |
| | UNIT (Liu et al., 2017) | x | x | ✓ | x |
| | MFM (Tsai et al., 2019) | ✓ | x | x | ✓ |
| **Ours** | | ✓ | ✓ | ✓ | ✓ |

Figure 2: [Left] The four criteria for multi-modal generative models: (a) latent factorisation (b) coherent joint generation (c) coherent cross generation and (d) synergy. [Right] A characterisation of recent work that explores multiple modalities, including our own, in terms of the specified criteria. See § 2 for further details.

With this in mind, when designing algorithms that imitate the human learning process, we seek a generative model that is able to jointly embed and generate multi-modal observations, which learn concepts by association of multiple modalities and feedback from their reconstructions. While the variational autoencoder (VAE) (Kingma and Welling, 2014) fits such a description well, truly capturing the range of behaviour and abilities exhibited by humans from multi-modal observation requires enforcing particular characteristics on the framework itself. Although there have been a range of approaches that broadly tackle the issue of multi-modal generative modelling (c.f. § 2), they fall short of expressing a more complete range of expected behaviour in this setting. We hence posit four criteria that a multi-modal generative model should satisfy (c.f. Figure 2[left]):

**Latent Factorisation:** the latent space implicitly factors into subspaces that capture the *shared* and *private* aspects of the given modalities. This aspect is important from the perspective of downstream tasks, where better decomposed representations (Lipton, 2018; Mathieu et al., 2019) are more amenable for use on a wider variety of tasks.

**Coherent Joint Generation:** generations in different modalities stemming from the same latent value exhibit coherence in terms of the shared aspects of the latent. For example, in the schematic in Figure 1, this could manifest through the generated image and description always matching semantically—that is, the description is true of the image.

**Coherent Cross Generation:** the model can generate data in one modality conditioned on data observed in a different modality, such that the underlying commonality between them is preserved. Again taking Figure 1 as an example, for a given description, one should be able to generate images that are semantically consistent with the description, and vice versa.

**Synergy:** the quality of the generative model for any single modality is improved through representations learnt across multi-modal observations, as opposed to just the single modality itself. That is, observing both the image and description should lead to more specificity in generation of the images (and descriptions) than when taken alone.

To this end, we propose the MMVAE, a multi-modal VAE that uses a mixture of experts (MoE) variational posterior over the individual modalities to learn a multi-modal generative model that satisfies the above four criteria. While it shares some characteristics with the most recent work on multi-modal generative models (cf. § 2), it nonetheless differs from them in two important ways. First, we are interested in situations where 1) observations across multiple modalities are always presented during training; 2) trained model can handle missing modalities at test time. Second, our experiments use many-to-many multi-modal mapping scenario, which provides a greater challenge than the typically used one-to-one image ↔ image (colourisation or edge/outline detection) and image ↔ attribute transformations (image classification). To the best of our knowledge, we are also the first to explore image ↔ language transformation under the multi-modal VAE setting. Figure 2[right] summarises relevant work (cf. § 2) and identifies if they satisfy our proposed criteria.

## 2 Related Work

**Cross-modal generation** Prior approaches to generative modelling with multi-modal data have predominantly only targeted cross-modal generation. Given data from two domains $x_1$ and $x_2$,

they learn the conditional generative model $p(\boldsymbol{x}_1 \mid \boldsymbol{x}_2)$, where the conditioning modality $\boldsymbol{x}_2$ and generation modality $\boldsymbol{x}_1$ are typically *not* interchangeable. This is commonly seen in conditional VAE methods for attribute→image or image→caption generation (Pandey and Dukkipati, 2017; Pu et al., 2016; Sohn et al., 2015; Yan et al., 2016), as well as generative adversarial network (GAN)-based models for cross domain image-to-image translation (Ledig et al., 2017; Li and Wand, 2016; Liu et al., 2019; Taigman et al., 2017; Wang and Gupta, 2016). In recent years, there have been a few approaches involving both VAEs and GANs that enable cross-modal generation *both* ways (Wang et al., 2016; Zhu et al., 2017a,b), but ignore learning a common embedding between them, instead treating the different cross-generations as independent but composable transforms. Curiously some GAN cross-generation models appear to avoid learning abstractions of data, choosing instead to hide the actual input directly in the high-frequency components of the output (Chu et al., 2017).

**Domain adaptation**  The related sub-field of domain adaptation explores learning joint embeddings of multi-modal observations that generalise across the different modalities for both classification (Long et al., 2015, 2016; Tzeng et al., 2014) and generation (Bousmalis et al., 2016) tasks. And regarding approaches that go beyond just cross-modal generation to models that learn common embedding or projection spaces, Ngiam et al. (2011) were the first to employ an autoencoder-based architecture to learn joint representation between modalities, using their RBM-CCA model. Silberer and Lapata (2014) followed this with a stacked autoencoder architecture to jointly embed the visual and textual representations of nouns. Tian and Engel (2019) considers an intermediary embedding space to transform between *independent* VAEs, applying constraints on the closeness of embeddings and their individual classification performance on labels.

**Joint models**  Yet another class of approaches attempt to explicitly model the joint distribution over latents and data. Suzuki et al. (2017) introduced the joint multimodal VAE (JMVAE) that learns shared representation with joint encoder $q_\Phi(\boldsymbol{z} \mid \boldsymbol{x}_1, \boldsymbol{x}_2)$. To handle missing data at test time, two unimodal encoders $q_\Phi(\boldsymbol{z} \mid \boldsymbol{x}_1)$ and $q_\Phi(\boldsymbol{z} \mid \boldsymbol{x}_2)$ are trained to match $q_\Phi(\boldsymbol{z} \mid \boldsymbol{x}_1, \boldsymbol{x}_2)$ with a KL constraint between them. Vedantam et al. (2018)'s multimodal VAE TELBO (triple ELBO) also deals with missing data at test time by explicitly defining multimodal and unimodal inference networks. However, differing from the JMVAE, they facilitate the convergence between the unimodal encoding distributions and the joint distribution using a two-step training regime that first fits the joint encoding, and subsequently fits the unimodal encodings holding the joint fixed. Tsai et al. (2019) propose MFM (multimodal factorisation model), which also explicitly defines a joint network $q_\Phi(\boldsymbol{z} \mid \boldsymbol{x}_{1:M})$ on top of the unimodal encoders, seeking to infer missing modalities using the observed modalities.

We argue that these approaches are less than ideal, as they typically only target *one* of the proposed criteria (e.g. (one-way) cross-generation), often require additional modelling components and inference steps (JMVAE, TELBO, MFM), ignore the latent representation structures induced and largely only target observations within a particular domain, typically vision $\leftrightarrow$ vision (Liu et al., 2017). More recently, Wu and Goodman (2018) introduced the MVAE, a marked improvement over previous approaches, proposing to model the joint posterior as a product of experts (PoE) over the marginal posteriors, enabling cross-modal generation at test time *without* requiring additional inference networks and multi-stage training regimes. While already a significant step forward, we observe that the PoE factorisation does not appear to be practically suited for multi-modal learning, likely due to the precision miscalibration of experts. See § 3 for more detailed explanation. We also observe that latent-variable mixture models have previously been applied to generative models targeting multi-modal topic-modelling (Barnard et al., 2003; Blei and Jordan, 2003). Although differing from our formulation in many ways, these approaches nonetheless indicate the suitability of mixture models for learning from multi-modal data.

## 3   Methods

**Background**  We employ a VAE (Kingma and Welling, 2014) to learn a multi-modal generative model over modalities $m = 1, \ldots, M$ of the form $p_\Theta(\boldsymbol{z}, \boldsymbol{x}_{1:M}) = p(\boldsymbol{z}) \prod_{m=1}^{M} p_{\theta_m}(\boldsymbol{x}_m \mid \boldsymbol{z})$, with the likelihoods $p_{\theta_m}(x_m \mid \boldsymbol{z})$ parametrised by deep neural networks (decoders) with parameters $\Theta = \{\theta_1, \ldots, \theta_M\}$. The objective of training VAEs is to maximise the marginal likelihood of the data $p_\Theta(\boldsymbol{x}_{1:M})$. However, computing the evidence is intractable as it requires knowledge of the true joint posterior $p_\Theta(\boldsymbol{z} \mid \boldsymbol{x}_{1:M})$. To tackle this, we approximate the true *unknown* posterior by a variational posterior $q_\Phi(\boldsymbol{z} \mid \boldsymbol{x}_{1:M})$, which now allows optimising an evidence lower bound (ELBO)

through stochastic gradient descent (SGD), with ELBO defined as

$$\mathcal{L}_{\mathrm{ELBO}}(\boldsymbol{x}_{1:M}) = \mathbb{E}_{\boldsymbol{z} \sim q_{\Phi}(\boldsymbol{z}|\boldsymbol{x}_{1:M})} \left[ \log \frac{p_{\Theta}(\boldsymbol{z}, \boldsymbol{x}_{1:M})}{q_{\Phi}(\boldsymbol{z} \mid \boldsymbol{x}_{1:M})} \right] \tag{1}$$

The importance weighted autoencoder (IWAE) (Burda et al., 2015) computes a tighter lower bound through appropriate weighting of a multi-sample estimator, as

$$\mathcal{L}_{\mathrm{IWAE}}(\boldsymbol{x}_{1:M}) = \mathbb{E}_{\boldsymbol{z}^{1:K} \sim q_{\Phi}(\boldsymbol{z}|\boldsymbol{x}_{1:M})} \left[ \log \sum_{k=1}^{K} \frac{1}{K} \frac{p_{\Theta}(\boldsymbol{z}^k, \boldsymbol{x}_{1:M})}{q_{\Phi}(\boldsymbol{z}^k \mid \boldsymbol{x}_{1:M})} \right] \tag{2}$$

Beyond the targetting of a tighter bound, we further prefer the IWAE estimator since the variational posteriors it estimates tend to have higher entropy (see Appendix D). This is actually beneficial in the multi-modal learning scenario, as each posterior $q_{\phi_m}(\boldsymbol{z} \mid x_m)$ is encouraged to assign high probability to regions *beyond* just those which characterise its own modality $m$.

**The mixture of experts (MoE) joint variational posteriors**   Given the objective, a crucial question however remains: how should we learn the variational joint posterior $q_{\Phi}(\boldsymbol{z} \mid \boldsymbol{x}_{1:M})$?

One immediately obvious approach is to train one single encoder network that takes all modalities $\boldsymbol{x}_{1:M}$ as input to explicitly parametrise the joint posterior. However, as described in § 2, this approach requires all modalities to be presented at all times, thus making cross-modal generation difficult. We propose to factorise the joint variational posterior as a combination of unimodal posteriors, using a mixture of experts (MoE), i.e. $q_{\Phi}(\boldsymbol{z} \mid \boldsymbol{x}_{1:M}) = \sum_m \alpha_m \cdot q_{\phi_m}(\boldsymbol{z} \mid \boldsymbol{x}_m)$, where $\alpha_m = 1/M$, assuming the different modalities are of comparable complexity (as per motivation in § 1).

*MoE vs. PoE*   An alternative choice of factorising the joint variational posterior is as a product of experts (PoE), i.e. $q_{\Phi}(\boldsymbol{z} \mid \boldsymbol{x}_{1:M}) = \prod_m q_{\phi_m}(\boldsymbol{z} \mid \boldsymbol{x}_m)$, as seen in MVAE (Wu and Goodman, 2018). When employing PoE, each expert holds the power of veto—in the sense that the joint distribution will have low density for a given set of observations if just one of the marginal posteriors has low density. In the case of Gaussian experts, as is typically assumed[2], experts with greater precision will have more influence over the combined prediction than experts with lower precision. When the precisions are miscalibrated, as likely in learning with SGD due to difference in complexity of input modalities or initialisation conditions, overconfident predictions by one expert—implying a potentially biased mean prediction overall—can be detrimental to the whole model. This can be undesirable for learning factored latent representations across modalities. By contrast, MoE does not suffer from potentially overconfident experts, since it effectively takes a vote amongst the experts, and spreads its density over all the individual experts. This characteristic makes them better-suited to latent factorisation, being sensitive to information across all the individual modalities. Moreover Wu and Goodman (2018) noted that PoE does not work well when observations across all modalities are always presented during training, requiring artificial subsampling of the observations to ensure that the individual modalities are learnt faithfully. As evidence, we show empirically in § 4 that the PoE factorisation does not satisfy all the criteria we outline in § 1.

**The MoE-multimodal VAE (MMVAE) objective**   With the MoE joint posterior, we can extend the $\mathcal{L}_{\mathrm{IWAE}}$ in (2) to multiple modalities by employing stratified sampling (Robert and Casella, 2013) to average over $M$ modalities:

$$\mathcal{L}_{\mathrm{IWAE}}^{\mathrm{MoE}}(\boldsymbol{x}_{1:M}) = \frac{1}{M} \sum_{m=1}^{M} \mathbb{E}_{\boldsymbol{z}_m^{1:K} \sim q_{\phi_m}(\boldsymbol{z}|\boldsymbol{x}_m)} \left[ \log \frac{1}{K} \sum_{k=1}^{K} \frac{p_{\Theta}(\boldsymbol{z}_m^k, \boldsymbol{x}_{1:M})}{q_{\Phi}(\boldsymbol{z}_m^k \mid \boldsymbol{x}_{1:M})} \right], \tag{3}$$

which has the effect of weighing the gradients of samples from different modalities *equally* while still estimating tight bounds for each individual term. Note that although an even tighter bound can be computed by weighting the contribution of each modality differently, in proportion to its contribution to the marginal likelihood, doing so can lead to undesirable modality dominance similar to that in the PoE case. See Appendix A for further details and results.

It is easy to show that $\mathcal{L}_{\text{IWAE}}^{\text{MoE}}(\boldsymbol{x}_{1:M})$ is still a tighter lower bound than the standard $M$-modality ELBO using linearity of expectations, as

$$\mathcal{L}_{\text{ELBO}}(\boldsymbol{x}_{1:M}) = \mathbb{E}_{q_\Phi(\boldsymbol{z}|\boldsymbol{x}_{1:M})}\left[\log \frac{p_\Theta(\boldsymbol{z}, \boldsymbol{x}_{1:M})}{q_\Phi(\boldsymbol{z} \mid \boldsymbol{x}_{1:M})}\right] = \frac{1}{M}\sum_{m=1}^{M} \mathbb{E}_{\boldsymbol{z}_m \sim q_{\phi_m}(\boldsymbol{z}|\boldsymbol{x}_m)}\left[\log \frac{p_\Theta(\boldsymbol{z}_m, \boldsymbol{x}_{1:M})}{q_\Phi(\boldsymbol{z}_m \mid \boldsymbol{x}_{1:M})}\right]$$

$$\leq \frac{1}{M}\sum_{m=1}^{M} \mathbb{E}_{\boldsymbol{z}_m^{1:K} \sim q_{\phi_m}(\boldsymbol{z}|\boldsymbol{x}_m)}\left[\log \frac{1}{K}\sum_{k=1}^{K} \frac{p_\Theta(\boldsymbol{z}_m^k, \boldsymbol{x}_{1:M})}{q_\Phi(\boldsymbol{z}_m^k \mid \boldsymbol{x}_{1:M})}\right] = \mathcal{L}_{\text{IWAE}}^{\text{MoE}}(\boldsymbol{x}_{1:M}).$$

In actually computing the gradients for the objectives in Equations (3) and (5) we employ the DReG IWAE estimator of Tucker et al. (2019), avoiding issues with the quality of the estimator for large $K$ as discovered by Rainforth et al. (2018) (cf. Appendix C).

From a computational perspective, the MoE objectives incur some overhead over the PoE objective, due to the fact that each modality provides samples from its own encoding distribution $q_{\phi_m}(\boldsymbol{z} \mid \boldsymbol{x}_m)$ to be evaluated with the joint generative model $p_\Theta(\boldsymbol{z}, \boldsymbol{x}_{1:M})$, needing $M^2$ passes over the respective decoders in total. The real cost of such added complexity however can be minimal since a) the number of modalities one can simultaneously process is typically quite small, and b) the additional computation can be efficiently vectorised. However, if this cost should be deemed prohibitively large, one can in fact trade off the tightness of the estimator for linear time complexity in the number of modalities $M$, employing a multi-modal importance sampling scheme on the standard ELBO. We discuss this in further detail in appendix B.

## 4 Experiments

To evaluate our model, we constructed two multi-modal scenarios to conduct experiments on. The first experiment involves many-to-many image $\leftrightarrow$ image transforms on matching digits between the MNIST and street-view house numbers (SVHN) datasets. This experiment was designed to separate perceptual complexity (i.e. color, style, size) from conceptual complexity (i.e. digits) using relatively simple image domains. The second experiment involves a highly challenging image $\leftrightarrow$ language task on the Caltech-UCSD Birds (CUB) dataset—more complicated than the typical image $\leftrightarrow$ attribute transformations employed in prior work. We choose this dataset as it matches our original motivation in tackling multi-modal perception in a similar manner to how humans perceive and learn about the world. For each of these experiments, we provide both qualitative and quantitative analyses of the extent to which our model satisfies the four proposed criteria—which, to reiterate, are i) implicit latent decomposition, ii) coherent joint generation over all modalities, iii) coherent cross-generation across individual modalities, and iv) improved model learning for individual modalities through multi-modal integration. Source code for all models and experiments is available at https://github.com/iffsid/mmvae.

### 4.1 Common Details

Across experiments, we employ Laplace priors and posteriors, constraining their scaling across the $D$ dimensions to sum to $D$. These design choices better encourage the learning of axis-aligned representations by breaking the rotationally-invariant nature of the standard isotropic Gaussian prior (Mathieu et al., 2019). For learning, we use the Adam optimiser (Kingma and Ba, 2014) with AMSGrad (Reddi et al., 2018), with a learning rate of 0.001. Details of the architectures used are provided in Appendix F. All numerical results were averaged over 5 independently trained models. Data and pre-trained models from our experiments are also available at https://github.com/iffsid/mmvae.

### 4.2 MNIST-SVHN

**Dataset:** As mentioned before, we design this experiment in order to probe *conceptual* complexity separate from *perceptual* complexity. We construct a dataset of pairs of MNIST and SVHN such that each pair depicts the same digit class. Each instance of a digit class (in either dataset) is randomly paired with 20 instances of the same digit class from the other dataset. As shown in Figure 3, although the data domains are fairly well known, effectively capturing the digit class can be a challenging task due to the variety of styles and colours presented across both datasets. Here, we use CNNs for SVHN and MLPs for MNIST, with a $20d$ latent space.

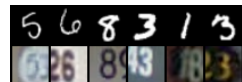

Figure 3: Example data

MMVAE (ours)  MVAE (Wu and Goodman, 2018)

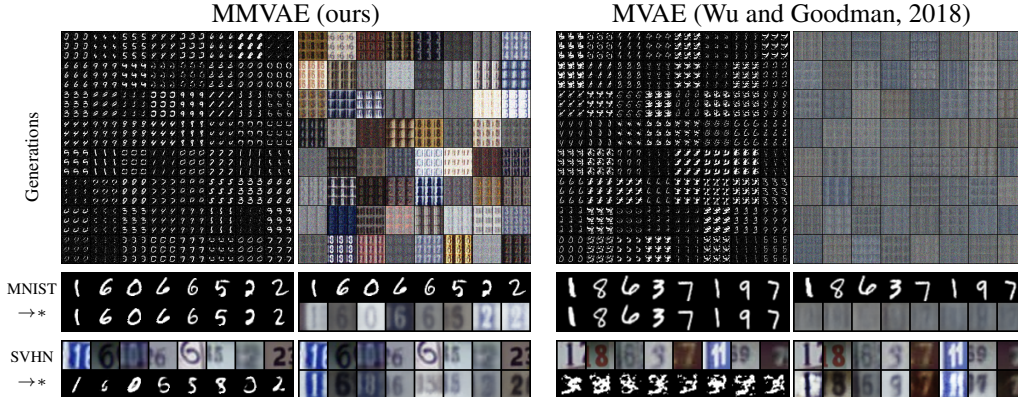

Figure 4: Qualitative evaluation of both our MMVAE model and MVAE from Wu and Goodman (2018). Generations (top row) for each modality. Note both the quality of generations and the extent to which corresponding generations in MNIST and SVHN match on digits for MMVAE vs. MVAE, satisfying the coherent generation criteria. Reconstructions and cross-generations for MNIST (middle row) and SVHN (bottom row). Again note the extent to which cross generations capture the underlying digit effectively for MMVAE.

**Qualitative Results:** Figure 4 shows a qualitative comparison between MMVAE trained with the MOE objective in (3) against the MVAE model of Wu and Goodman (2018). The MVAE was trained using the authors' publicly available code[3], following their recommended training regime. We generate from the model $p_\Theta(z, x_{1:M})$ by taking $R = 64$ samples from the prior $z \sim p(z)$, each of which is used to take $N = 9$ samples from the likelihood of each modality $m$ as $x_m \sim p_\Theta(x_m \mid z)$. Note the quality of the MMVAE model, both at coherent joint generation (top row)—where corresponding samples for the same $z$ match in their digits—and at coherent cross-generation (middle and bottom rows). To show that it is truly the MOE factorisation that impacts the learning—rather than our particular choice of model architecture or the IWAE objective—we explore performance on *only adopting* MOE, directly in the codebase of Wu and Goodman (2018), keeping all other aspects fixed, in Appendix E. Results indicate MVAE with the MOE posterior does appear to do better, especially at cross-modal generation, than the POE.

We subsequently analyse the structure of the latent space by traversing each dimension independently as shown in Figure 5. Here, for each modality $m$, we encode datapoint $x_m$ through its respective encoder $q_{\phi_m}(z \mid x_m)$ to obtain the mean embedding $\mu_m$. Then, we perturb the embedding value along each dimension $\mu_m^d$ linearly in the range $(\mu_m^d - 5\sigma_0^d, \mu_m^d + 5\sigma_0^d)$, where $\sigma_0^d$ is the *learnt* standard deviation for dimension $d$ in the prior $p(z)$. Note the extent to which particular dimensions affect only a single modality, whereas other dimensions affect both, indicating a degree of latent factorisation. Also shown is a plot of the per-dimension Kullback-Leibler divergence (KL) between each posterior and the prior, as well as the symmetric KL between the two posteriors, to indicate which dimensions encode information from which posterior, if any.

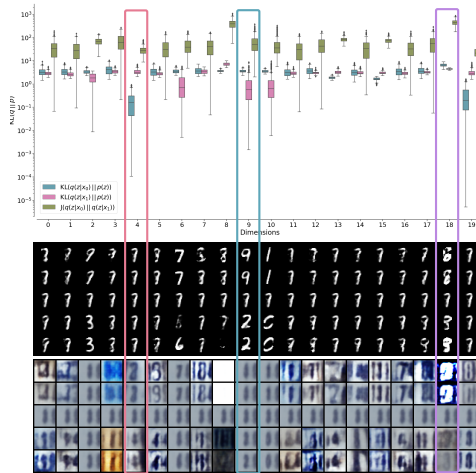

Figure 5: Per-dimension latent traversals for a pair of datapoints indicating dimensions that affect only SVHN, only MNIST, and both MNIST & SVHN.

**Quantitative Results:** To quantify the extent to which the latent spaces factorises from multi-modal observations, we employ a simple linear classifier on the latent representations as we have no *a-priori* reason to believe that the representations factorise in an axis-aligned manner. If a linear digit classifier can extract the digit information from the shared latent space, it strongly indicates the presence of a linear subspace that has factored as desired. We train digit classifiers for a) MMVAE, b) MVAE, with posterior either from single-modality inputs or multi-modality inputs, and c) single-VAE that takes input from one modality only, plotting results in Table 1. Comparing the results of the first

column to the last in Table 1, we find MMVAE's latent space provides significantly better accuracy over the single-VAE. For the MVAE, due to its PoE formulation, it appears that the MNIST modality dominates, obtaining high accuracy for MNIST digit classification (95.7%) but low for SVHN (9.10%). When given *both* inputs, accuracy for SVHN improves significantly while that for MNIST decreases slightly. Note that the accuracy for MVAE is higher when only the MNIST data is presented compared to when both modalities are available, indicating that the presence of the extra modality does not further inform the model on the classification of digits.

Table 1: Digit classification accuracy (%) of latent variables in different models.

|  | **MMVAE** | MVAE (single) | MVAE (both) | single-VAE |
|---|---|---|---|---|
| MNIST | 91.3 | 95.7 | 94.9 | 85.3 |
| SVHN | 68.0 | 9.1 | 90.1 | 20.7 |

We also quantify the *coherence* of joint generations and cross-modal generations. To do so, we employ off-the-shelf digit classifiers of the original MNIST and SVHN datasets on the generation results, and compute a) for joint generation, how often the digits of generations in two modalities match, and b) for cross-modal generation, how often the digit generated in one modality matches its input from another modality. Results in Table 2 indicate that for joint generation, the classifiers predict the same digit class 42.1% of time, and for cross-generation, 86.4% (MNIST →SVHN) and 69.1% (SVHN → MNIST). Computing these metrics for MVAE yields accuracy close to chance, suggesting that the coherence between modalities is not quite preserved when considering generation.

Table 2: Probability of digit matching (%) for joint and cross generation.

|  | Joint | Cross (M→S) | Cross (S→M) |
|---|---|---|---|
| **MMVAE** | 42.1 | 86.4 | 69.1 |
| MVAE | 12.7 | 9.5 | 9.3 |

We finally compute the marginal likelihoods[4] of the joint generative model $p_\Theta(\boldsymbol{x}_{1:M})$, and each of the individual generative models $p_{\theta_m}(x_m)$ using both the *joint* variational posterior $q_\Phi(\boldsymbol{z} \mid \boldsymbol{x}_{1:M})$ and the *single* variational posterior $q_{\phi_m}(\boldsymbol{z} \mid x_m)$ as shown in Table 3.

Table 3: Evaluating the different log likelihoods for different arrangements of MNIST and SVHN.

|  |  | $\log p(\boldsymbol{x}_m, \boldsymbol{x}_n)$ | $\log p(\boldsymbol{x}_m \mid \boldsymbol{x}_m, \boldsymbol{x}_n)$ | $\log p(\boldsymbol{x}_m \mid \boldsymbol{x}_m)$ | $\log p(\boldsymbol{x}_m \mid \boldsymbol{x}_n)$ |
|---|---|---|---|---|---|
| $m$ = MNIST, | **MMVAE** | 6261.40 | 868.76 | 868.37 | 628.31 |
| $n$ = SVHN | MVAE | 2961.80 | −176.68 | −107.46 | −778.20 |
| $m$ = SVHN, | **MMVAE** | 6261.40 | 3441.01 | 3441.01 | 2337.56 |
| $n$ = MNIST | MVAE | 2961.80 | 3395.12 | 3536.86 | −12747.50 |

We observe that MMVAE model yields higher likelihoods, consistent with employing the IWAE estimator. Interestingly, we observe that $p(\boldsymbol{x}_m \mid \boldsymbol{x}_m, \boldsymbol{x}_n) \geq p(\boldsymbol{x}_m \mid \boldsymbol{x}_m)$ for MMVAE, whereas for MVAE we consistently find $p(\boldsymbol{x}_m \mid \boldsymbol{x}_m, \boldsymbol{x}_n) < p(\boldsymbol{x}_m \mid \boldsymbol{x}_m)$. This serves to highlights that the MMVAE model is able to effectively utilise information jointly across multiple modalities, which the MVAE model potentially suffers from overdominant encoders and an ill-suited sub-sampled training scheme to accommodate data always present across modalities at train time.

## 4.3 CUB Image-Captions

**Dataset:** Encouraged by the results from our previous experiment, we consider a multi-modal experiment more in line with our original motivation. We employ the images and captions from Caltech-UCSD Birds (CUB) dataset (Wah et al., 2011), containing 11,788 photos of birds in natural scenes, each annotated with 10 fine-grained captions describing the bird's appearance characteristics collected through Amazon Mechanical Turk (AMT). As shown in Figure 6, the images are quite detailed and descriptions fairly complex, involving the composition of various attributes. For the image data, rather than generating directly in image space, we

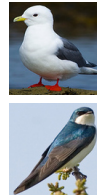

the bird has a white body, black wings, and webbed orange feet

a blue bird with gray primaries and secondaries and white breast and throat

Figure 6: Example data

instead generate in the feature space of a pre-trained ResNet-101 (He et al., 2016), in order to avoid issues with blurry generations for complex image data (Zhao et al., 2017). For generations and

reconstructions, we simply perform a nearest-neighbour lookup in feature space using Euclidean distance on the generated or reconstructed feature. For the language data, we employ a CNN encoder and decoder following Kalchbrenner et al. (2014); Massiceti et al. (2018b); Pham et al. (2016), learning an embedding for words in the process. We use 128-dimensional latents with a Laplace likelihood on image features and a Categorical likelihood for captions.

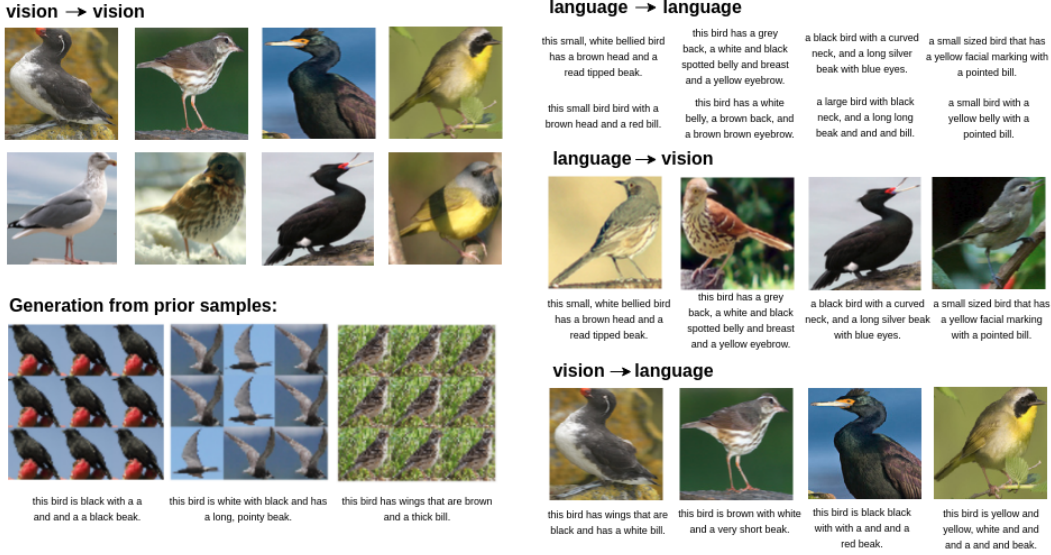

Figure 7: Qualitative evaluation of the MMVAE model on the CUB data, showing reconstruction in the individual modalities (top rows), cross generation (middle and bottom rows on right), and joint generation (bottom row on left). More qualitative examples can be found in Appendix G.

**Qualitative Results:** Figure 7 shows qualitative results for the MMVAE model trained with the MOE objective in (3) on the CUB data. We generate from the model as before, with $R = 4$, and $N_1 = 9, N_2 = 1$. Interestingly, even for such a complicated dataset, we see joint generation align quite well with descriptions largely in line with the image for a range of different attributes, and cross generation where the descriptions again match the image quite well and vice versa. An interesting avenue to explore for future directions in this experiment would be the incorporation of more complex generative models for images potentially incorporating the use of GANs to improve the quality of generated output, and observing its effect on the joint modelling. We also generate results for MVAE on this dataset, where we observe that the image modality dominates and resulting in poor language reconstruction and cross-modal generation. See Appendix H for examples and further details.

**Quantitative Results:** We evaluate the coherence of our generation by calculating the correlation between the jointly and cross-generated image-caption pair. We do so by employing Canonical Correlation Analysis (CCA) following the observation of its effectiveness as a baseline for language and vision tasks by Massiceti et al. (2018a). Here, given paired observations $\{x_1 \in \mathbb{R}^{n_1}, x_2 \in \mathbb{R}^{n_2}\}$, CCA learns projections $W_1 \in \mathbb{R}^{n_1 \times k}$ and $W_2 \in \mathbb{R}^{n_2 \times k}$ that maximise the correlation between projected variables $W_1^T x_1$ and $W_2^T x_2$. With this formulation, the correlation between any new pair of observations $\{\tilde{x}_1, \tilde{x}_2\}$ can be computed as the cosine distance between the mean-centered projected variables, i.e.

$$\text{corr}(\tilde{x}_1, \tilde{x}_2) = \frac{\phi(\tilde{x}_1)^T \phi(\tilde{x}_2)}{||\phi(\tilde{x}_1)||_2 ||\phi(\tilde{x}_2)||_2} \tag{4}$$

where $\phi(\tilde{x}_n) = W_n^T \tilde{x}_n - \text{avg}(W_n^T x_n)$.

We prepare the dataset for CCA by pre-processing both modalities using feature extractors. For images, similar to training, we use the off-the-shelf ResNet-101 to generate feature vector of dimension $2048$-$d$; for captions, we fit a FastText model on all sentences in the training set, projecting each word onto a $300$-$d$ vector (Bojanowski et al., 2017). The representation for each caption is then acquired by aggregating the embedding of all words in the sentence (here we simply take the average). To compute the correlation between the generated images and captions, we first compute the projection matrix $W$ for each modality using the training set of CUB Image-Captions, then perform

CCA using (4), on i) jointly generated image-sentence pairs, taking average over 1000 examples, and ii) image-sentence or sentence-image pair of cross generation, taking an average over the entire test set. Results are as shown in Table 4.

Table 4: Correlation of Image (I)-Sentence (S) pair for joint and cross generation.

| | Joint | Cross (I →S) | Cross (S →I) | Ground Truth |
|---|---|---|---|---|
| **MMVAE** | 0.263 | 0.104 | 0.135 | 0.273 |
| MVAE | −0.095 | 0.011 | −0.013 | |

Table 4 shows that the average correlation of joint generation of our model is 0.263; This value is only slightly lower than the average correlation of the data itself (0.273 in Table 4), which demonstrates the high coherence between the jointly generated image-caption pairs. For cross-generation, the correlation between *input* images and *generated* captions is slightly lower than that of *input* caption and *generated* image, evaluated at 0.104 and 0.135 respectively.

In comparison, the MVAE model appears to provide (marginally) negative correlation for the jointly generated pairs and sentence → image cross generation pairs. Notably, the correlation for image → sentence generation is much higher than sentence → image (0.011 and -0.013 respectively). Observing the qualitative results for MVAE in Appendix H also shows that any outputs generated from images as input is more expressive than those from the language inputs. These findings indicate that the model places more weight on the image modality than language for the factorisation of joint posterior, once again indicating an overdominant encoder, providing empirical evidence for our analysis of the PoE's potential bias towards stronger experts in § 3.

## 5   Conclusion

In this paper, we explore multi-modal generative models, characterising successful learning of such models as the fulfillment of four specific criteria: i) implicit latent decomposition into shared and private subspaces, ii) coherent simultaneous joint generation over all modalities, iii) coherent cross-generation between individual modalities, and iv) improved model learning for the individual modalities as a consequence of having observed data from different modalities. Satisfying these goals enables more useful and generalisable representations for downstream tasks such as classification, by capturing the abstract relationship between the modalities. To this end, we propose a variational mixture of experts (MoE) autoencoder framework that allows us to achieve these criteria, in contrast to prior work which primarily target just the cross-modal generation and improved model learning aspects. We compare and contrast our MMVAE model against the state-of-the-art product of experts (PoE) model of Wu and Goodman (2018) and demonstrate that we outperform it at satisfying these four criteria. We evaluate our model on two challenging datasets that capture both image ↔ image and image ↔ language transformations, showing appealing results across these tasks.

**Acknowledgements**

YS, NS, and PHST were supported by the ERC grant ERC-2012-AdG 321162-HELIOS, EPSRC grant Seebibyte EP/M013774/1 and EPSRC/MURI grant EP/N019474/1, with further support from the Royal Academy of Engineering and FiveAI. YS was additionally supported by Remarkdip through their PhD Scholarship Programme. BP is supported by the Alan Turing Institute under the EPSRC grant EP/N510129/1.

## Footnotes

[2]Training PoE models in general can be intractable (Hinton, 2002) due to the required normalisation, but becomes analytic when the experts are Gaussian.

[3]https://github.com/mhw32/multimodal-vae-public

[4]We compute a 1000-sample estimate using (5) here.

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
