[Supplementary Material]

# Appendix: Variational Mixture-of-Experts Autoencoders for Multi-Modal Deep Generative Models

## A  Tighter lower bound

With the MoE joint posterior, an alternative way of extending the $\mathcal{L}_{\text{IWAE}}$ in (2) to $M$-modalities is by employing stratified sampling (Robert and Casella, 2013)—to take $K$ samples from the joint posterior, we first sample a modality $m$, then take $L = K/M$ samples from the corresponding marginal variational posterior $q_{\phi_m}(z \mid x_m)$, and repeat the process $M$ times. Formally,

$$\underline{\mathcal{L}}_{\text{IWAE}}^{\text{MoE}}(x_{1:M}) = \mathbb{E}_{\substack{z_1^{1:L} \sim q_{\phi_1}(z|x_1) \\ \vdots \\ z_M^{1:L} \sim q_{\phi_M}(z|x_M)}} \left[ \log \frac{1}{M} \sum_{m=1}^{M} \frac{1}{L} \sum_{l=1}^{L} \frac{p_\Theta(z_m^l, x_{1:M})}{q_\Phi(z_m^l \mid x_{1:M})} \right]. \tag{5}$$

Applying Jensen's inequality, we see that (5) is a tighter lower bound than (3); that is, $\mathcal{L}_{\text{IWAE}}^{\text{MoE}}(x_{1:M}) \leq \underline{\mathcal{L}}_{\text{IWAE}}^{\text{MoE}}(x_{1:M}) \leq \log p_\Theta(x_{1:M})$; however, it can adversely affect the ability to perform cross-modal generation. To see this, consider the form of the gradient estimator of this objective with respect to the variational posterior parameters $\Phi$ (Burda et al., 2015; Cremer et al., 2017),

$$\nabla_\Phi \underline{\mathcal{L}}_{\text{IWAE}}^{\text{MoE}}(x_{1:M}) = \mathbb{E}_{\epsilon^{1:K} \sim p(\epsilon)} \left[ \sum_k \bar{w}^k \nabla_\Phi \log w^k \right], \tag{6}$$

where $K = ML$, $z^k = g(\epsilon^k, \Phi)$ (reparameterised), $w^k = \frac{p_\Theta(z^k, x_{1:M})}{q_\Phi(z^k|x_{1:M})}$, and $\bar{w}^k = \frac{w^k}{\sum_{j=1}^{K} w^j}$, indicating that the gradient weights samples by their relative importance $\bar{w}^k$. Two different samples $z^{k_1}$, and $z^{k_2}$ coming from different modalities (encoders) can have their gradients weighed differently from one another, leading to situations where the joint variational posterior collapses to one of the experts in the mixture. Figure 8 shows empirical evidence of this happening, comparing performance under Equations (3) and (5) for the same data.

Figure 8: Comparing performance for MNIST-SVHN between the $\mathcal{L}_{\text{IWAE}}^{\text{MoE}}(x_{1:M})$ and $\underline{\mathcal{L}}_{\text{IWAE}}^{\text{MoE}}(x_{1:M})$ objectives. Compared to the original objective (LHS), the cross generation of $\widehat{\mathcal{L}}_{\text{IWAE}}^{\text{MoE}}(x_{1:M})$ struggles to match digits between the modalities, especially in the SVHN → MNIST case. Coherence for joint generation is also worse, with no recognisable matching-up of digits between the two modalities. Similar to Table 2, we evaluate the coherence of cross and joint generation by computing the accuracy of digit predictions between the two modalities in Table 5. Note the drop in performance for the $\underline{\mathcal{L}}_{\text{IWAE}}^{\text{MoE}}(x_{1:M})$ objective reflecting the qualitative results.

| Objective | Joint | Cross (M →S) | Cross (S →M) |
|---|---|---|---|
| $\mathcal{L}_{\text{IWAE}}^{\text{MoE}}(x_{1:M})$ | 42.1 | 86.4 | 69.1 |
| $\underline{\mathcal{L}}_{\text{IWAE}}^{\text{MoE}}(x_{1:M})$ | 24.2 | 74.6 | 15.6 |

Table 5: Probability of digit matching (%) for joint and cross generation.

## B Multi-Modal Importance-Sampled ELBO

Consider the ELBO, with the basic MOE variational posterior,

$$\mathcal{L}_{\text{ELBO}}(\boldsymbol{x}_{1:M}) = \frac{1}{M} \sum_{m=1}^{M} \mathbb{E}_{z \sim q_{\phi_m}(\boldsymbol{z}|\boldsymbol{x}_m)} \left[ \log \frac{p_\Theta(\boldsymbol{z}, \boldsymbol{x}_{1:M})}{q_\Phi(\boldsymbol{z} \mid \boldsymbol{x}_{1:M})} \right].$$

The term corresponding to a particular modality $i$ is given as

$$\mathbb{E}_{\boldsymbol{z}_i \sim q_{\phi_i}(\boldsymbol{z}|\boldsymbol{x}_i)} \left[ \log \frac{p_\Theta(\boldsymbol{z}_i, \boldsymbol{x}_1, \ldots, \boldsymbol{x}_i, \ldots, \boldsymbol{x}_M)}{q_\Phi(\boldsymbol{z}_i \mid \boldsymbol{x}_{1:M})} \right]$$

$$= \mathbb{E}_{\boldsymbol{z}_i \sim q_{\phi_i}(\boldsymbol{z}|\boldsymbol{x}_i)} \left[ \log \frac{p_\Theta(\boldsymbol{z}_i, \boldsymbol{x}_i)}{q_\Phi(\boldsymbol{z}_i \mid \boldsymbol{x}_{1:M})} \right] + \sum_{\substack{j=1 \\ j \neq i}}^{M} \mathbb{E}_{\boldsymbol{z}_i \sim q_{\phi_i}(\boldsymbol{z}|\boldsymbol{x}_i)} \left[ \log p_{\theta_j}(\boldsymbol{x}_j \mid \boldsymbol{z}_i) \right]$$

$$= \mathbb{E}_{\boldsymbol{z}_i \sim q_{\phi_i}(\boldsymbol{z}|\boldsymbol{x}_i)} \left[ \log \frac{p_\Theta(\boldsymbol{z}_i, \boldsymbol{x}_i)}{q_\Phi(\boldsymbol{z}_i \mid \boldsymbol{x}_{1:M})} \right] + \sum_{\substack{j=1 \\ j \neq i}}^{M} \underbrace{\mathbb{E}_{\boldsymbol{z}_j \sim \bar{q}_{\phi_j}(\boldsymbol{z}|\boldsymbol{x}_j)} \left[ \frac{q_{\phi_i}(\boldsymbol{z}_j \mid \boldsymbol{x}_i)}{\bar{q}_{\phi_j}(\boldsymbol{z}_j \mid \boldsymbol{x}_j)} \log p_{\theta_j}(\boldsymbol{x}_j \mid \boldsymbol{z}_j) \right]}_{A_j}$$

where $\bar{q}$ does not propagate gradients (effectively issuing a `stop_gradient`), since all the likelihoods in the second term only have gradients with respect to $\phi_i, \theta_j, j \neq i$. Each term $A_j$ can be seen as an importance-sampled estimate of $\mathbb{E}_{\boldsymbol{z}_i \sim q_{\phi_i}(\boldsymbol{z}|\boldsymbol{x}_i)} \left[ \log p_{\theta_j}(\boldsymbol{x}_j \mid \boldsymbol{z}_i) \right]$ using that modality $j$'s own encoding distribution $q_{\phi_j}(\boldsymbol{z}_j \mid \boldsymbol{x}_j)$. This has two major benefits:

i) it allows the total objective to be computed with just a single pass over each encoder and decoder to make computation linear in $M$. To see this is true, note that one can effectively precompute the likelihoods of modality $j$ using samples from its own encoder, and simply weigh them by the appropriate ratio of variational posteriors where necessary.

ii) estimating the likelihood of modality $j$ using samples from modality $i$, where $i \neq j$, is a difficult ask—since observation $\boldsymbol{x}_i$ only carries partial information applicable for the reconstruction of $\boldsymbol{x}_j$. This multi-modal importance sampling sidesteps the issue by using samples from same modality $j$, with appropriate weighting. We would expect this to generally thus produce a lower-variance estimator, as it avoids potentially evaluating $\log p_{\theta_j}(\boldsymbol{x}_j \mid \boldsymbol{z}_i)$ for values $\boldsymbol{z}_i$ which yield very low likelihoods on $\boldsymbol{x}_j$. Moreover, since the denominators of the importance ratios cannot propagate gradients, the only way to improve the estimate is by maximising the density of samples from modality $j$ in the variational posterior for $i$ as $q_{\phi_i}(\boldsymbol{z}_j \mid \boldsymbol{x}_i)$, bringing the different components closer to each other.

## C The DReG Estimator

The standard gradient estimator of IWAE can have undesirably high variance (Rainforth et al., 2018; Roeder et al., 2017). To see this, we can expand (6) as:

$$\nabla_\Phi \mathbb{E}_{z^{1:K}} \left[ \log \left( \frac{1}{K} \sum_{k=1}^{K} w^k \right) \right] = \mathbb{E}_{\epsilon^{1:K}} \left[ \sum_{k=1}^{K} \frac{w^k}{\sum_{j=1}^{K} w^j} \left( -\frac{\partial}{\partial \Phi} \log q_\Phi(\boldsymbol{z}^k \mid \boldsymbol{x}) + \frac{\partial \log w^k}{\partial \boldsymbol{z}^k} \frac{d\boldsymbol{z}^k}{d\Phi} \right) \right] \tag{7}$$

Roeder et al. (2017) find that when $K > 1$, the first term within paranthesis in (7) need not be zero even when the approximate posterior matches true posterior everywhere, which can contribute to significant variance in the gradient estimator. To alleviate this, Tucker et al. (2019) re-apply the reparametrisation trick on it, yielding a doubly reparametrised gradient estimator (DReG):

$$\nabla_\Phi E_{z^{1:K}} \left[ \log \left( \frac{1}{K} \sum_{k=1}^{K} w^k \right) \right] = E_{\epsilon^{1:K}} \left[ \sum_{k=1}^{K} \left( \frac{w^k}{\sum_{j=1}^{K} w^j} \right)^2 \frac{\partial \log w^k}{\partial \boldsymbol{z}^k} \frac{d\boldsymbol{z}^k)}{d\Phi} \right] \tag{8}$$

We implement the estimator specified in (8) when performing gradient updates for any experiment involving the IWAE objective.

# D Higher entropy of IWAE variational posteriors

To see why the variational posteriors estimated by IWAE tend to have higher entropy than those by ELBO, it is beneficial to consider the objectives from a different perspective.

First, let's take a look at ELBO: maximising the standard ELBO indirectly minimises the KL between the variational and true posteriors, since

$$\log p_\Theta(\boldsymbol{x}_{1:M}) = \mathcal{L}_{\text{ELBO}}(\boldsymbol{x}_{1:M}) + \text{KL}(q_\Phi(\boldsymbol{z} \mid \boldsymbol{x}_{1:M}) \,\|\, p_\Theta(\boldsymbol{z} \mid \boldsymbol{x}_{1:M})).$$

Maximising the IWAE objective however, indirectly minimises the KL between *implicit* posteriors (Le et al., 2018) as

$$\log p_\Theta(\boldsymbol{x}_{1:M}) = \mathcal{L}_{\text{IWAE}}(\boldsymbol{x}_{1:M}) + \text{KL}(q_{\Phi^{IS}}(\boldsymbol{z} \mid \boldsymbol{x}_{1:M}) \,\|\, p_{\Theta^{IS}}(\boldsymbol{z} \mid \boldsymbol{x}_{1:M})),$$

where

$$p_{\Theta^{IS}}(\boldsymbol{z} \mid \boldsymbol{x}_{1:M}) = \frac{1}{K}\sum_k \frac{q_{\Phi^{IS}}(\boldsymbol{z} \mid \boldsymbol{x}_{1:M})}{q_\Phi(\boldsymbol{z}^k \mid \boldsymbol{x}_{1:M})} p_\Theta(\boldsymbol{z}^k \mid \boldsymbol{x}_{1:M}),$$

$$q_{\Phi^{IS}}(\boldsymbol{z} \mid \boldsymbol{x}_{1:M}) = \frac{1}{K}\prod_k q_\Phi(\boldsymbol{z}^k \mid \boldsymbol{x}_{1:M}),$$

leading to higher-entropy estimates of the variational posterior.

This suits the learning of multi-modal data as each modality's posterior attempts to explain *more* than just its own modality.

# E Qualitative results of MVAE implementation with MOE posterior

Figure 9: A comparison of POE vs. MOE in the MVAE codebae.

Here we show a qualitative evaluation in the MVAE codebase, minimally altered to use a MOE joint approximate posterior, with the results of the original POE-MVAE model as a comparison. We can see that MOE is able to generate recognisable MNIST digits from SVHN inputs (bottom row, column 3), while the original model fails completely at cross-modal generation. Although, do note that neither model performs well at coherence joint generation (top row).

# F Encoder and decoder architectures

| Encoder | Decoder |
|---|---|
| Input $\in \mathbb{R}^{1x28x28}$ | Input $\in \mathbb{R}^L$ |
| FC. 400 ReLU | FC. 400 ReLU |
| FC. $L$, FC. $L$ | FC. 1 x 28 x 28 Sigmoid |

(a) MNIST dataset.

| Encoder |
|---|
| Input $\in \mathbb{R}^{1x28x28}$ |
| 4x4 conv. 32 stride 2 pad 1 & ReLU |
| 4x4 conv. 64 stride 2 pad 1 & ReLU |
| 4x4 conv. 128 stride 2 pad 1 & ReLU |
| 4x4 conv. L stride 1 pad 0, 4x4 conv. L stride 1 pad 0 |

| Decoder |
|---|
| Input $\in \mathbb{R}^L$ |
| 4x4 upconv. 128 stride 1 pad 0 & ReLU |
| 4x4 upconv. 64 stride 2 pad 1 & ReLU |
| 4x4 upconv. 32 stride 2 pad 1 & ReLU |
| 4x4 upconv. 3 stride 2 pad 1 & Sigmoid |

(b) SVHN dataset.

| Encoder | Decoder |
|---|---|
| Input $\in \mathbb{R}^{2048}$ | Input $\in \mathbb{R}^L$ |
| FC. 1024 ELU | FC. 256 ELU |
| FC. 512 ELU | FC. 512 ELU |
| FC. 256 ELU | FC. 1024 ELU |
| FC. $L$, FC. $L$ | FC. 2048 |

(c) CUB image dataset.

| Encoder |
|---|
| Input $\in \mathbb{R}^{1590}$ |
| Word Emb. 256 |
| 4x4 conv. 32 stride 2 pad 1 & BatchNorm2d & ReLU |
| 4x4 conv. 64 stride 2 pad 1 & BatchNorm2d & ReLU |
| 4x4 conv. 128 stride 2 pad 1 & BatchNorm2d & ReLU |
| 1x4 conv. 256 stride 1x2 pad 0x1 & BatchNorm2d & ReLU |
| 1x4 conv. 512 stride 1x2 pad 0x1 & BatchNorm2d & ReLU |
| 4x4 conv. L stride 1 pad 0, 4x4 conv. L stride 1 pad 0 |

| Decoder |
|---|
| Input $\in \mathbb{R}^L$ |
| 4x4 upconv. 512 stride 1 pad 0 & ReLU |
| 1x4 upconv. 256 stride 1x2 pad 0x1 & BatchNorm2d & ReLU |
| 1x4 upconv. 128 stride 1x2 pad 0x1 & BatchNorm2d & ReLU |
| 4x4 upconv. 64 stride 2 pad 1 & BatchNorm2d & ReLU |
| 4x4 upconv. 32 stride 2 pad 1 & BatchNorm2d & ReLU |
| 4x4 upconv. 1 stride 2 pad 1 & ReLU |
| Word Emb.$^T$ 1590 |

(d) CUB-Language dataset.

Table 6: Encoder and decoder architectures.

# G  Qualitative results of MMVAE on CUB

In this section, we show some more qualitative results of our MMVAE model on CUB Image-Caption dataset.

Figure 10: Image reconstruction of MMVAE on CUB Image-Caption dataset. Top row: ground truth, bottom row: reconstruction.

[DATA]  ==> a small bird with a white belly and a grey wing and a short beak . <eos>
[RECON] ==> a small bird with a white belly and a brown , and a short beak. <eos>

[DATA]  ==> bird has brown body feathers , brown breast feathers , and brown beak . <eos>
[RECON] ==> bird has brown body feathers , brown breast feather , and black beak . <eos>

[DATA]  ==> this bird is brown with white and has a long , pointy beak . <eos>
[RECON] ==> this bird is black with white and has a long , pointy beak . <eos>

[DATA]  ==> this is an orange bird with a black wing and head . <eos>
[RECON] ==> this is a red bird with a wings and and beak . <eos>

[DATA]  ==> this bird has wings that are grey and has a yellow tail . <eos>
[RECON] ==> this bird has wings that are grey and has a yellow belly . <eos>

[DATA]  ==> this bird is yellow with black on its head and has a long , pointy beak . <eos>
[RECON] ==> this bird is yellow with grey on its head and has a long , pointy beak <eos>

[DATA]  ==> this bird has a white crown as well as a long black bill . <eos>
[RECON] ==> this bird has a white crown , black primaries , and a white . <eos>

[DATA]  ==> this bird has a speckled belly and breast with a short pointy bill . <eos>
[RECON] ==> this bird has a speckled belly and breast with a short pointy bill . <eos>

[DATA]  ==> this bird is white with brown and has a very short beak . <eos>
[RECON] ==> this bird is white with belly and and a a short beak . <eos>

[DATA]  ==> this bird has black feathers and a thick black bill . <eos>
[RECON] ==> this bird is completely black with a black orange bill . <eos>

[DATA]  ==> this particular bird has a belly that is white and black . <eos>
[RECON] ==> this particular bird has a belly that is black and white . <eos>

[DATA]  ==> the bird has a grey belly and white coverts on the wingbars . <eos>
[RECON] ==> this bird has a black crown and black with and black bill . <eos>

Figure 11: Caption reconstruction of MMVAE on CUB Image-Caption dataset.

[RECON] ==> this bird is grey with are and has a very short beak . <eos>

[RECON] ==> this is is black bird with a black and and a and a and beak . <eos>

[RECON] ==> this bird bird with long white neck and orange bill . <eos>

[RECON] ==> this small bird has a white belly, yellow breast and a black head . <eos>

[RECON] ==> this bird is bird with a white belly and a black bill . <eos>

[RECON] ==> this bird has wings that are brown and has a white belly . <eos>

[RECON] ==> this small bird has a bright body and and ,,, black and black head . <eos>

[RECON] ==> this bird has wings that are brown and a white belly . <eos>

[RECON] ==> this bird is brown and brown in color , with a stubby small beak . <eos>

[RECON] ==> this bird is brown with white and has a long, pointy beak . <eos>

[RECON] ==> this bird has wings that are grey and has a yellow belly . <eos>

[RECON] ==> a small bird with a brown body and brown beak . <eos>

[DATA] ==> a small sized bird that has a creamy belly with a short pointed bill . <eos>

[DATA] ==> small, mostly yellow bird, with brown, white, and black stripes on his wings and tail . <eos>

[DATA] ==> this bird is grey with black and has a long, pointy beak . <eos>

[DATA] ==> this small bird is mostly white with gray wings and tail as well as black stripes on its head . <eos>

[DATA] ==> this bird has a shiny all black body with a down pointed bill . <eos>

[DATA] ==> the bird is black in color, with a black beak . <eos>

[DATA] ==> the bird has a yellow crown, grey back and yellow coverts . <eos>

[DATA] ==> this bird has a grey body which fades to light brown on the breast . <eos>

[DATA] ==> this bird is red with black and has a very short beak . <eos>

[DATA] ==> this yellow bird has a black head and black primaries . <eos>

[DATA] ==> a light red colored bird, with black wings, and a sharp bill . <eos>

[DATA] ==> a bird with a small pointed bill, white eyebrows, and fluffy white and brown breast . <eos>

[GEN] ==> this bird is yellow and yellow, with a grey bill . <eos>

[GEN] ==> this small bird with a brown bill and white brown and a a of its . <eos>

[GEN] ==> this small and brown bird with a white belly . and a black bill . <eos>

[GEN] ==> this bird has wings that are black and has a long bill . <eos>

[GEN] ==> this bird has a red belly and belly and grey wings . <eos>

[GEN] ==> this bird is white with black and has a very short beak beak . <eos>

[GEN] ==> this small bird has yellow with black and and yellow belly . <eos>

[GEN] ==> this large bird with a and and a long bill . <eos>

[GEN] ==> this bird has a black crown, a long and black <eos>

[GEN] ==> this particular bird has a white belly and wings that are brown . <eos>

[GEN] ==> this bird has a black crown and a white and and white belly . <eos>

[GEN] ==> this bird is head black, with black yellow wings . <eos>

(a) Image → Caption          (b) Caption → Image          (c) Joint Generation

Figure 12: Cross generation (a, b) and joint generation from prior samples (c) of MMVAE on CUB Image-Caption dataset.

# H Qualitative results of Wu and Goodman (2018)'s MVAE on CUB

The qualitative results of MVAE on CUB Image-Caption dataset are as shown in Figure 13 and Figure 14. Note that similar to the MMVAE experiments, for the generation in the vision modality, we reconstruct the image features extracted from ResNet101 and perform nearest neighbour search to find the corresponding images.

**vision → vision**

**language → vision**

[DATA] ==> a small sized bird that has a cream belly and red head. <eos>

[DATA] ==> a bird with an orange beak, black head and neck with black wings. <eos>

[DATA] ==> this is a blue bird with black wings and a large pointy beak. <eos>

**language → language**

[DATA] ==> a small sized bird that has a cream belly and red head . <eos>
[RECON] ==> a medium sized bird that with a black , and a small . <eos>

[DATA] ==> a bird with an orange beak , black head and neck with black wings . <eos>
[RECON] ==> a bird with a white bill and white , , and and black beak . <eos>

[DATA] ==> this is a blue bird with black wings and a large pointy beak . <eos>
[RECON] ==> this is a white bird with black wings and a long orange beak . <eos>

[DATA] ==> this bird has wings that are grey with a white belly . <eos>
[RECON] ==> this bird has wings that are black with a white belly . <eos>

**vision → language**

[RECON] ==> this little bird has a yellow belly and breast with a brown crown and nape, and and and and wings. <eos>

[RECON] ==> this is a bird with a white belly, yellow that and and breast white, and a. <eos>

[RECON] ==> this colorful bird has a speckled that is breast breast green primaries slim <eos>

**Joint generation from prior samples:**

[GEN] ==> this bird has a white black with white white. <eos>

[GEN] ==> this is a white bird bird bird a with, white and and and wings. <eos>

[GEN] ==> this bird is brown in color, a a and beak and and and a and beak. <eos>

Figure 13: Qualitative results of MVAE on CUB Image-Caption dataset, including reconstruction (vision → vision, language → language), cross generation (vision → language, language → vision) and joint generation from prior samples.

**language, vision ——————→ language, vision**

[DATA] ==> this bird is brown with grey on its head and has a long , pointy beak . <eos>

[DATA] ==> a bird with long black wings , tail and white breast , the bill is short and black . <eos>

[DATA] ==> a small bird that 's primarily white , with yellow on top of its head with black over the eyes , yellow and black wings , and a beautiful <eos>

[DATA] ==> the bird has a brown back and a white belly along with a curved bill . <eos>

[DATA] ==> a small gray bird with white and dark gray wingbars and white breast . <eos>

[DATA] ==> this bird has wings that are black and has a white belly . <eos>

[DATA] ==> this bird has wings that are brown and has spots on them . <eos>

[DATA] ==> this brown bird has a white throat , orange bill , and a yellow breast and flank . <eos>

[RECON] ==> a small has a with with white a red over , and brown dark pointed secondaries <eos>

[RECON] ==> a large has with white with a white beak white throat and and , a white on and beak beak <eos>

[RECON] ==> this bird bird has throat with throat , face face , , , the , and and and and back , , white white white , body white black medium medium wings

[RECON] ==> a beautiful has a small with a a throat breast , a a dark white secondaries <eos>

[RECON] ==> this is a bird in with a white back white and medium white beak . <eos>

[RECON] ==> this bird has wings that are brown and , a big bill . <eos>

[RECON] ==> this bird has is has bird with wings with a red bill . <eos>

[RECON] ==> the bird has a a with with and beak and , secondaries and dark dark its wings beak. <eos>

Figure 14: Qualitative results of MVAE on CUB Image-Caption dataset. Here both modalities are given to reconstruct the inputs.

We can see from the results in Figure 13 that the vision modality dominates, with almost perfect image feature reconstruction; however, the performance in all other tasks are quite poor, especially when language is given as input: the language reconstruction fails to recover some of the key characteristics of the original description, replacing "small sized" with "medium sized", "blue bird" with "white bird" etc.; the language-vision cross generation suffers from mode collapse, generating exclusively the 2 images under the language → vision column in Figure 13 for any given caption.

The language generation both in reconstruction (language → language) and cross-modal generation (vision → language) fails to capture the key characteristics the original caption; neither are joint generation of image-caption pairs coherent,. No significant improvement in reconstruction quality can be observed when both modalities are given as input, as seen in Figure 14, with the language generation omitting/"making up" important characteristics of the bird images. Performing CCA analysis on these image-caption pairs gives a negative correlation of -0.00523 (averaged over the test set), suggesting low coherence of the generated multi-modal data.