[Reviews · NeurIPS 2019]

Reviewer 1



* Update after rebuttal: After reading the other reviews and the authors' response, I have decided to maintain my score. I appreciate the clarifications about the MNIST-SVHN protocol and the authors' willingness to add pixel-level generation and MVAE-on-CUB results as requested. On the other hand, my questions about the likelihood results remained unaddressed, and I still feel it is misleading that the qualitative visual results in Fig. 7 (via feature look-up) are presented as 'reconstruction' and 'generation'. (There may have been a misunderstanding about what I meant by 'missing modalities'—when individual samples may not have all modalities available, rather than a modality being unobserved for the entire dataset. I realise my wording was not very clear, so I am disregarding this point.) Summary ------------ The authors posit that a good generative model for multi-modal data should be able to achieve latent factorisation, coherent joint and cross generation, and synergy between modalities. They propose a deep generative model based on variational autoencoders (VAE) and mixture-of-experts (MoE), which satisfies these criteria and sidesteps some limitations of existing models. One branch for each modality and a shared latent space. Originality ------------- The formulation of the desiderata for multi-modal generative modelling data feels quite original. The related work section is detailed and fairly comprehensive, clearly delineating the authors' contribution. In particular, it differs from the closest related method (MVAE; Wu & Goodman, 2018) in significant and non-trivial aspects. Quality --------- - It is unclear why the authors limit their scope to cases wherein all modalities are fully available during training (lines 65-66), as this could be a deal-breaker for some applications. Could the training objective not be adapted somehow to allow missing modalities? There should be some discussion of this limitation and/or of potential solutions. - There is no discussion of the likelihood results at the end of Sec. 4.3. What is the message here? What should we have expected to see? Do these numbers confirm the authors' hypotheses? If the expected difference was between the third and fourth columns ('p(xm|xm,xn)' vs. 'p(xm|xm)'), then the effect was minuscule... Also, the sentence describing the calculation of these results is quite confusing (lines 276-279), and this is not helped by the table's lack of a title and caption---which makes it even harder to interpret the results. - The experiment with the Caltech-UCSD Birds (CUB) dataset are quite preliminary. The authors apply their model on the outputs of a pre-trained deep feature extractor, on the grounds of avoiding blurriness of generated images. They then report 'generated samples' as the nearest-neighbour training images in this feature space. This is a really understated limitation, and for example the visual results shown in Fig. 7 can be very misleading to less attentive readers. - Although I am not deeply familiar with recent NLP literature, the caption generation results feel fairly unconvincing in terms of text quality. There are also several inconsistencies in language->language and image->language generation, so the authors' claim of 'quality and coherence' (Fig. 7 caption) of the sampled captions appears quite subjective. It would be useful to show many more examples of these five tasks (e.g. in the supplement) to give readers a better picture of what the model is able to learn from this data. - Why is there no comparison with the MVAE baseline (Wu & Goodman, 2018) on CUB? Clarity -------- - Generally the exposition is very clear and pleasurable to read. - I believe it could be helpful for readers if the authors added to lines 112-123 a brief intuitive description of their MoE formulation. For example, my understanding of the mixture-of-experts is that evidence can come from *either* modality, rather than the product-of-experts' assumption that *all* modalities should agree. - The initial description of the MMVAE objective in lines 124-136 felt a bit convoluted to me, and I am not sure if it is currently adding much value in the main text. In fact, the jump from IWAE straight to the explanation starting from line 137 seems a lot more intuitive to begin with. Since the authors seemed pressed for space, I would actually suggest moving the initial formulation and surrounding discussion to the supplement. - It would be interesting to briefly justify the choice of factorised Laplace distributions for priors, posteriors and likelihoods, and to clarify how the constraint on the scales is enforced during training. - In the interest of squeezing in more content, the authors seem to have tampered with some spacing elements: the abstract's lateral margins and the gaps above section headings are visibly reduced. Some additional space could be spared e.g. by deleting the superfluous Sec. 4.1 heading and rephrasing to get rid of dangling short lines of text (e.g. 257, 318, 334). - Consider increasing the spacing around inline figures (Figs. 1, 3, 5, and 6), as it currently feels very tight and makes the surrounding text a bit harder to read. - The authors could clarify that MMVAE refers to multi-modal MoE VAE (or MoE multi-modal VAE?), to better distinguish it from Wu & Goodman (2018)'s multi-modal VAE (MVAE). - The multiple likelihood samples (N=9) in Fig. 4 do not seem to provide any additional information or insight, especially given the heavy speckle noise in the generated images. Here it might be clearer to display just the predicted mean for each sampled latent code. - The description of the quantitative analysis protocol for MNIST-SVHN (lines 258-267) needs some further clarifications. Is it classifying digits? Are the SVHN and MNIST accuracies computed by feeding through only data from the corresponding modality? To which model does the 94.5% on line 267 refer, and what are the implications of this result? - Minor fixes: * Line 18: 'lack *of* explicit labels' * Line 19: 'provided' => 'provide' * Line 25: '[...] them (Yildirim, 2014).' * Lines 41, 43, 63, 72, 145: 'c.f.' => 'cf.' * Line 64: 'end-to-end' * Line 166: 'targetted' => 'targeted' * Line 205: '..' => '.' * Line 214: 'recap' sounds a bit informal * Line 238: 'generative' => 'generate' * Line 293: 'euclidean' => 'Euclidean' * Ref. Kalchbrenner et al. (2014): author list is messed up - The authors cite no fewer than 17 arXiv preprints, most of which were subsequently published and have been available for a while. Consider searching for the updated references and citing the published works instead. Significance ---------------- The authors formalisation of criteria for multi-modal generative modelling is inspiring. The model and inference algorithm could also raise interest and invite extensions. While there are several issues with the evaluation on the more complex CUB dataset, the MNIST-SVHN experiments are interesting and informative, and suggest improvements over the MVAE baseline.

Reviewer 2



The paper is well organized and easy to understand. The use of MoE as approximate posterior is proved to be effective across several different tasks. Weakness: 1. Although the four criteria (proposed by the author of this paper) for multi-modal generative models seem reasonable, they are not intrinsic generic criteria. Therefore, the argument that previous works fail for certain criteria is not strong. 2. Tabular data (seeing each attribute dimension as a modality) is another popular form of multi-modal data. It would interesting, although not necessary, to see how this model works for tabular data.

Reviewer 3



****************************Originality**************************** Strengths: - The paper does a nice job of discussing the various desiderata of successful multimodal models in the introduction, nicely decomposing these into generative and discriminative objectives. Weaknesses: - The paper is not that original given the amount of work in learning multimodal generative models: — For example, from the perspective of the model, the paper builds on top of the work by Wu and Goodman (2018) except that they learn a mixture of experts rather than a product of experts variational posterior. — In addition, from the perspective of the 4 desirable attributes for multimodal learning that the authors mention in the introduction, it seems very similar to the motivation in the paper by Tsai et al. Learning Factorized Multimodal Representations, ICLR 2019, which also proposed a multimodal factorized deep generative model that performs well for discriminative and generative tasks as well as in the presence of missing modalities. The authors should have cited and compared with this paper. ****************************Quality**************************** Strengths: - The experimental results are nice. The paper claims that their MMVAE modal fulfills all four criteria including (1) latent variables that decompose into shared and private subspaces, (2) be able to generate data across all modalities, (3) be able to generate data across individual modalities, and (4) improve discriminative performance in each modality by leveraging related data from other modalities. Let's look at each of these 4 in detail: — (1) Yes, their model does indeed learn factorized variables which can be shown by good conditional generation on MNIST+SVHN dataset. — (2) Yes, joint generation (which I assume to mean generation from a single modality) is performed on vision -> vision and language -> language for CUB, — (3) Yes, conditional generation can be performed on CUB via language -> vision and vice versa. Weaknesses: - (continuing on whether the model does indeed achieve the 4 properties that the authors describe) — (3 continued) However, it is unclear how significant the performance is for both 2) and 3) since the authors report no comparisons with existing generative models, even simple ones such as a conditional VAE from language to vision. In other words, what if I forgo with the complicated MoE VAE, and all the components of the proposed model, and simply use a conditional VAE from language to vision. There are many ablation studies that are missing from the paper especially since the model is so complicated. — (4) The authors have not seemed to perform extensive experiments for this criteria since they only report the performance of a simple linear classifier on top of the latent variables. There has been much work in learning discriminative models for multimodal data involving aligning or fusing language and vision spaces. Just to name a few involving language and vision: - Multimodal Compact Bilinear Pooling for Visual Question Answering and Visual Grounding, EMNLP 2016 - DeViSE: A Deep Visual-Semantic Embedding Model, NeurIPS 2013 Therefore, it is important to justify why I should use this MMVAE model when there is a lot of existing work on fusing multimodal data for prediction. ****************************Clarity**************************** Strengths: - The paper is generally clear. I particularly liked the introduction of the paper especially motivation Figures 1 and 2. Figure 2 is particularly informative given what we know about multimodal data and multimodal information. - The table in Figure 2 nicely summarizes some of the existing works in multimodal learning and whether they fulfill the 4 criteria that the authors have pointed out to be important. Weaknesses: - Given the authors' great job in setting up the paper via Figure 1, Figure 2, and the introduction, I was rather disappointed that section 2 did not continue on this clear flow. To begin, a model diagram/schematic at the beginning of section 2 would have helped a lot. Ideally, such a model diagram could closely resemble Figure 2 where you have already set up a nice 'Venn Diagram' of multimodal information. Given this, your model basically assigns latent variables to each of the information overlapping spaces as well as arrows (neural network layers) as the inference and generation path from the variables to observed data. Showing such a detailed model diagram in an 'expanded' or 'more detailed' version of Figure 2 would be extremely helpful in understanding the notation (which there are a lot), how MMVAE accomplishes all 4 properties, as well as the inference and generation paths in MMVAE. - Unfortunately, the table in Figure 2 it is not super complete given the amount of work that has been done in latent factorization (e.g. Learning Factorized Multimodal Representations, ICLR 2019) and purely discriminative multimodal fusion (i.e. point d on synergy) - There are a few typos and stylistic issues: 1. line 18: "Given the lack explicit labels available” -> “Given the lack of explicit labels available” 2. line 19: “can provided important” -> “can provide important” 3. line 25: “between (Yildirim, 2014) them” -> “between them (Yildirim, 2014)” 4. and so on… ****************************Significance**************************** Strengths: - This paper will likely be a nice addition to the current models we have for processing multimodal data, especially since the results are quite interesting. - The paper did a commendable job in attempting to perform experiments to justify the 4 properties they outlined in the introduction. - I can see future practitioners using the variational MoE layers for encoding multimodal data, especially when there is missing multimodal data. Weaknesses: - That being said, there are some important concerns especially regarding the utility of the model as compared to existing work. In particular, there are some statements in the model description where it would be nice to have some experimental results in order to convince the reader that this model compares favorably with existing work: 1. line 113: You set \alpha_m uniformly to be 1/M which implies that the contributions from all modalities are the same. However, works in multimodal fusion have shown that dynamically weighting the modalities is quite important because 1) modalities might contain noise or uncertain information, 2) different modalities contribute differently to the prediction (e.g. in a video when a speaker is not saying anything then their visual behaviors are more indicative than their speech or language behaviors). Recent works therefore study, for example, gated attentions (e.g. Gated-Attention Architectures for Task-Oriented Language Grounding, AAAI 2018 or Multimodal Sentiment Analysis with Word-level Fusion and Reinforcement Learning, ICMI 2017) to learn these weights. How does your model compare to this line of related work, and can your model be modified to take advantage of these fusion methods? 2. line 145-146: "We prefer the IWAE objective over the standard ELBO objective not just for the fact that it estimates a tighter bound, but also for the properties of the posterior when computing the multi-sample estimate." -> Do you have experimental results that back this up? How significant is the difference? 3. line 157-158: "needing M^2 passes over the respective decoders in total" -> Do you have experimental runtimes to show that this is not a significant overhead? The number of modalities is quite small (2 or 3), but when the decoders are large recurrent of deconvolutional layers then this could be costly. ****************************Post Rebuttal**************************** The author response addressed some of my concerns regarding novelty but I am still inclined to keep my score since I do not believe that the paper is substantially improving over (Wu and Goodmann, 2018) and (Tsai et al, 2019). The clarity of writing can be improved in some parts and I hope that the authors would make these changes. Regarding the quality of generation, it is definitely not close to SOTA language models such as GPT-2 but I would still give the authors credit since generation is not their main goal, but rather one of their 4 defined goals to measure the quality of multimodal representation learning.

[Author Response · NeurIPS 2019]

We thank the reviewers for their detailed comments and are encouraged by their positive appraisals. We take on board all
comments regarding typos, spacing, and citations [**R1** & **R3**] and will fix these appropriately in the updated manuscript.

[**R1** ... *fully-available modalities* ... ] Although we are primarily interested in the cognitive-science motivation (§1)
for all modalities being available, our formulation *can* indeed deal with missing modalities by virtue of the MoE
properties—effectively amounting to assuming $\alpha_m = 0$ for missing modality $m$. Note however that doing so (as with
[1] using the PoE), can raise questions of *consistency*. Constructing a joint encoder with a missing modality does not
change the target—it simply is 'another' importance distribution—but, the decoder missing a modality changes the
actual model being targetted since it is not the full generative model $p_\theta(z, x_{1:M})$ anymore! It is unclear what the right
approach in this case ought to be; and in conjunction with our motivation, is why we do not explore this setting here.

[**R1** ... *experimental protocol* ... *results* ... ] We will include a more thorough discussion of the quantitative analysis for
the experiments and the results. We do indeed classify digits, and use inputs from MNIST or SVHN to classify for that
corresponding modality (67% & 87.3% resp.). The 94.5% corresponds to the MVAE of [1] when *both* modalities are
given as inputs—the implication being that it hedges its abilities entirely on MNIST (94.42% when given just MNIST
input), largely ignoring SVHN (10.5% when given only SVHN)—a consequence of the PoE formulation.

[**R1** ... *more convincing experiment* ... *CUB* ... ] Although our focus in this work is not on pixel-level generation,
(as also noted by **R3**), we will include pixel-level-generation results on a downscaled version of CUB images in the
supplementary as requested. We omitted results for MVAE on CUB because they were quite poor, and results on
MNIST-SVHN served sufficiently to highlight its shortcomings. We will however include these results in the updated
manuscript for completeness.

[**R2** ... *not intrinsic generic criteria* ... ] We believe these criteria are still quite general in that they are typically easy to
verify experimentally—the procedures in §4 aim to provide a blueprint for how to do so, and we will make this protocol
more explicit in the updated manuscript.

[**R2** *Tabular data* ... ] Deep generative models for tabular data often focus on learning proper co-occurrence structure
between different attributes, so it may be difficult to apply this method directly (which would treat each attribute as
conditionally independent); although we agree this is an interesting avenue for future work.

[**R3** ... *not that original* ... *Tsai et al.* ... ] The main question across prior approaches [incl. 2,3] has been the
formulation of inference. While different choices have yielded different capabilities (c.f. Fig 2, right) we believe, and
show, that MoE posteriors are better. Characterising our contribution as 'not that original' does not do us justice. Thank
you for the citation to Tsai et al. We note that although the motivation in latent factorisation is shared, the means are
quite different—we do not explicitly structure the latent space (as originally seen in [4]), allowing the factorisation to
be captured automatically (c.f. Fig 5). We also show coherent joint generation from the unconditional prior.

[**R3** ... *no comparisons* ... *conditional VAE* ... ] Our focus here is on learning *joint* generative models that additionally
allows for (computationally) cheap conditional generation. Of course, we agree that if one wanted only to learn a
conditional model, say from language to vision, then explicitly targeting that capability alone would be useful; but
that isn't our goal—and such would not be an apples-to-apples comparison. The value of learning a joint model lies
in the ability to learn, in an unsupervised manner, the abstract commonalities in the observed data—in being able to
unconditionally generate (from the prior $p(z)$) data in different modalities that are related in the same way the observed
data was—for MNIST-SVHN the digit (c.f. Fig 4 left) , and for CUB, more nuanced notions of language grounding (c.f.
Fig 7 bottom left). The motivation comes from human perception (§1) and is not something that conditional models do.

[**R3** ... *ablations* ... ] In comparisons against [1] we have since performed additional ablations to find out which aspect
of our formulation provides the performance win—since we differ from [1] in both form of posterior (MoE vs PoE) *and*
the estimator used (DReG vs ELBO)—and can show that the MoE formulation is the critical component.

[**R3** ... *simple linear classifier* ... ] As stated in the manuscript (l258-l262) the linear classifier is used to quantify how
well-separated the information is in the latent space, not simply an attempt to perform state of the art classification.

[**R3** ... $\alpha_m$ *dynamic* ... ] This is indeed true—and can be handled by the DReG estimator. Note that Eq (1) computes
importance weights across both modality and samples from each modality, which is similar to the dynamic weighting in
multimodal fusion methods. We explicitly disallow importance weighting across modalities since different encoders
can contribute to a joint representation to different degrees, which we avoid in order to explore Criteria 1 (§1). However,
we will include an example in the supplement to showcase the dynamic weighting enabled by Eq (1).

[**R3** ... *IWAE/ELBO* ... $M^2$ ... *multiple runs* ... ] See [5,6] for the tightness and posterior properties of IWAE; see
Appendix A1 for linear-in-M objective. We will update the manuscript with statistics from runs across multiple seeds.

[1] Wu & Goodman, 2018  [2] Ngiam et al., 2019  [3] Blei & Jordan, 2003  [4] Bousmalis et al., 2016  [5] Burda et al., 2016
[6] Le et al., 2018


[Meta-Review · NeurIPS 2019]

All reviewers agree on the quality of this submission.